# Predicting Vertical Ground Reaction Forces in Running from the Sound of Footsteps

**DOI:** 10.3390/s22249640

**Published:** 2022-12-08

**Authors:** Anderson Souza Oliveira, Cristina-Ioana Pirscoveanu, John Rasmussen

**Affiliations:** 1Department of Materials and Production, Aalborg University, DK-9220 Aalborg East, Denmark; 2Department of Health Science and Technology, Aalborg University, DK-9220 Aalborg East, Denmark

**Keywords:** footsteps, running, sound, ground reaction forces, machine learning

## Abstract

From the point of view of measurement, footstep sounds represent a simple, wearable and inexpensive sensing opportunity to assess running biomechanical parameters. Therefore, the aim of this study was to investigate whether the sounds of footsteps can be used to predict the vertical ground reaction force profiles during running. Thirty-seven recreational runners performed overground running, and their sounds of footsteps were recorded from four microphones, while the vertical ground reaction force was recorded using a force plate. We generated nine different combinations of microphone data, ranging from individual recordings up to all four microphones combined. We trained machine learning models using these microphone combinations and predicted the ground reaction force profiles by a leave-one-out approach on the subject level. There were no significant differences in the prediction accuracy between the different microphone combinations (*p* < 0.05). Moreover, the machine learning model was able to predict the ground reaction force profiles with a mean Pearson correlation coefficient of 0.99 (range 0.79–0.999), mean relative root-mean-square error of 9.96% (range 2–23%) and mean accuracy to define rearfoot or forefoot strike of 77%. Our results demonstrate the feasibility of using the sounds of footsteps in combination with machine learning algorithms based on Fourier transforms to predict the ground reaction force curves. The results are encouraging in terms of the opportunity to create wearable technology to assess the ground reaction force profiles for runners in the interests of injury prevention and performance optimization.

## 1. Introduction

In the fable, *De Tre Bukkene Bruse*, whose first written record in 1840 is attributed to Norwegian folklorist Peter Christen Asbjørnsen, a troll preys on goats crossing a bridge to reach their pastures. From his hideout under the bridge, the troll assesses the size of the goats and the cost-benefit of the potential hunt by the sound of their footsteps. The fable builds on the intuition that, given experience with the surface (the bridge), the sound of a footstep is characteristic of the body weight, i.e., the development of the ground reaction force (GRF) in the stance phase of the gait. Despite this old and popular notion, the precise relationship between sound and the GRF has barely been investigated. It is plausible that the sound’s causal relationship to the GRF is elusive due to its dependency on complex physics phenomena such as stick-slip friction, sound reflection from the environment, material cracking and debris dynamics. However, the combination of recorded sounds and the GRF with machine learning techniques may allow for a phenomenological identification of a useful relationship between sound and force.

For runners, inexpensive wearable sensors based on inertial measurement units are available to detect accelerations, angular velocities and other kinematics data. However, assessing kinetic variables such as the GRF through wearable measurements is not similarly accessible. Running kinetic parameters extracted from the GRF are highly relevant for the assessment of running economy and performance, as well as injury prevention. The most accessible alternative is plantar pressure insoles, which are expensive and are delicate devices with a limited lifespan when used for running [1,2]. Therefore, alternative methods to indirectly access the GRF through regression methods have been established. Previous studies have predicted the GRF through machine/deep learning algorithms using several different data sources, such as plantar pressure and running slope [3], lower body kinematics [1,2], uniaxial accelerometers [3] and even hydrocell sensors [4]. In addition, other studies have focused on predicting specific features usually extracted from the vertical GRF (vGRF), such as the impact peak [1] and vertical instantaneous loading rate [4]. It is noteworthy that some of these prediction studies still require the use of expensive and complex recording methods, warranting the need for alternative methods using inexpensive sensors.

Running is an exercise widely known for the cumulative vertical loading that might induce acute and chronic lower limb injuries such as tibial stress fractures [5,6], tendinopathies and other musculoskeletal disorders at the lower limb and trunk [7,8,9,10]. Multiple studies have shown that assessing the changes in impact loading during running can be useful for preventing and/or reducing injury incidence accessible through variables such as vertical tibial acceleration and the vertical instantaneous loading rates [6,8,11]. Greater instantaneous impact loading may generate louder sounds of footsteps, which have been previously shown to be moderately correlated to vertical loading rates and peak propulsion forces [12,13]. The sounds of footsteps have been used in gait retraining to reduce instantaneous impact loading during running [7,14] and can be used to classify different running styles [15]. Moreover, muscle fatigue increases the loading rate [16,17], which can be strongly correlated with increases in peak footstep sounds in fatiguing conditions [13]. These studies present compelling evidence illustrating the relevance of sound waves for the investigation of running biomechanics. The acquisition of sound waves is relatively inexpensive and can be implemented wirelessly on footwear or as skin sensors or may even be recorded through mobile phones. Therefore, using the sounds of footsteps to predict/reconstruct the vGRF and their relevant running parameters could be highly relevant for runners.

The aim of this study was to investigate the feasibility of using the sounds of footsteps acquired from multiple microphones for the reconstruction/prediction of the vGRF time series using a machine learning model. Multiple predictions were conducted using a single microphone, pairs of microphones or all the microphones simultaneously. We firstly hypothesized that the machine learning model would be able to predict the vGRF time series with a moderate-to-high accuracy from the sounds, allowing for low relative errors when comparing traditional biomechanical parameters between the real and predicted vGRF. Moreover, we hypothesized that the vGRF prediction quality would be superior when using multiple sound sources.

Section 2 describes the experimental setup, machine learning method, data processing and statistical analysis. Section 3 presents the results, and Section 4 and Section 5 discuss the findings and the perspectives for an inexpensive and wearable sensing technology for the assessment of vertical ground reaction forces.

## 2. Materials and Methods

### 2.1. Participants

Forty-four recreational runners (8 females and 36 males, age: 26 ± 4 years, weight: 77 ± 11 kg, height 178 ± 9 cm, BMI 24 ± 2) volunteered to participate in this study. They all had a self-reported moderate-to-high level of physical activity (3269 ± 1543 MET), according to the International Physical Activity Questionnaire, or a minimum three years of running experience and 14 ± 9 km weekly mileage. During the testing or within the past 6 months prior, all participants were healthy, and no history of lower extremity musculoskeletal injury or running-related injuries was disclosed. Verbal and written informed consent before inclusion was provided by all runners. Furthermore, the experimental methods were carried out according to the relevant guidelines and regulations approved by the Ethical Committee of North Jutland (Region Nordjylland).

### 2.2. Experimental Setup

All participants were asked to perform overground running at their preferred running speed. Prior to testing, the runners had a 5-min warm-up consisting of walking lunges, skipping, leg swings and run-throughs [14], followed by a 5–10 min familiarization to the running track. During the warm-up, their preferred running speed was established. The running track was a 20-m indoor-obround running track (2-m running turns and 8-m sprinting lines) with a force platform and four surrounding microphones located in the middle of the first sprinting line (Figure 1) [18]. The microphones’ location did not interfere with the runners’ sprinting line trajectory.

Every participant was asked to run continuously for three minutes around the track at their preferred running speed for a total of three sets with a 3-min rest period in between. The runners were asked to strike with their right foot in the middle of the force plate. Furthermore, their running speed was continuously monitored on a trial-by-trial basis by one researcher, who recorded the time (and subsequently the speed) during a 3-m sector of the track that contained the force plate in the middle. The speed monitoring allowed for running speed feedback to be provided to the runners, in case their speed varied ±10% of the pre-established value in order for the runner to adjust accordingly in the next lap. A standard running shoe (Nike Air Pegasus) was used throughout the experiment.

### 2.3. Equipment

The track had a caption point which consisted of: (1) a floor-embedded force plate (AMTI Optima Gen 5, Watertown, MA, USA) with a sampling frequency of 1000 Hz that provided 3D forces and moments, (2) an 8-camera motion capture system (Qualisys 1.0, Qualisys Oqus, Göteborg, Sweden) with a sampling frequency of 200 Hz and (3) four dual-powered Røde NTG-2 shotgun microphones (Røde, Sydney, Australia), which recorded sounds sampled at 44.1 kHz using a sound card (Focusrite, Scarlet 18i8, High Wycombe, UK) connected to a custom-made Matlab Simulink script for sound acquisition [18].

The motion capture system was used to record the position of eight retro-reflexive markers (16 mm diameter). Four of the markers were fitted equidistantly onto a headband to allow for the calculation of running speed. The other four markers were positioned on the running shoe in the corresponding anatomical landmarks of the right calcaneus, fifth metatarsus and second toe in order to calculate the ankle initial contact in the sagittal plane angle. Furthermore, the motion capture system was used to synchronize the capture of the running kinetic and kinematic data. Moreover, the sound recordings were synchronized with the kinematic/kinetic data using an external trigger signal that started the sound recordings and was labeled as an event in the kinematic/kinetic data set [18].

The microphones were positioned at a 45°-degree angle from the floor, at a 10 cm distance and 15 cm height from the corners of the force plate. This positioning aimed to record the running sounds from the center of the force plate, following the previous literature [12,18]. Considering the foot placement on the force platform during running, microphone 1 (Mic1) was located left-posterior; microphone 2 (Mic2) was located anterior-left; microphone 3 (Mic3) was located anterior-right; and microphone (Mic4) was located posterior-right. The running sounds were converted from voltages to decibels as described elsewhere [13].

### 2.4. Kinetics, Kinematics and Sound Data Processing

Force data were filtered using a fourth-order Butterworth low-pass filter (60 Hz cut-off frequency) and normalized to the participants’ body weight. The running stance period was defined when the vertical force was above 20 N during foot contact to the force platform. The participants’ running speeds were calculated within a range of one meter before and after stepping onto the force plate from the mean displacement of the four markers positioned on the participants’ head through a headband for each successful running cycle. The foot contact angle (FCA) was determined from the two-dimensional ankle dorsiflexion angle, defined as the angle of the ankle at initial contact subtracted from the ankle dorsiflexion angle at mid-stance [12,13]. Running style was defined using the FCA to determine rearfoot strike (FCA ≥ 8°), midfoot strike (−1.6° < FCA < 8°) and forefoot strike (FCA ≤ −1.6°) [19]. The sound data from each microphone were initially resampled to 20,000 Hz, band-pass filtered (first-order Butterworth) using a band-pass range of 5 to 20,000 Hz. Subsequently, the sound data were full-wave rectified and low-pass filtered at 200 Hz to create a linear envelope of the sound profile. The final sound envelopes were segmented into running strides using the segmentation events from the vGRF.

### 2.5. Machine Learning

Prediction of ground reaction forces from microphone signals was performed by a machine learning algorithm initially developed to predict kinematic running patterns [20]. Briefly, the algorithm Fourier-transforms the input training data time signals (measured vGRF and sound wave signals) and uses the resulting coefficients as data. An extensive investigation of the Fourier transforms for kinematic data [21] previously established that six terms in the series sufficiently represented the signal. However, the need to capture the impact peak accurately in the vGRF signals and the low-pass-filtered sound variations led to the decision to use 15 terms for both signals following a visual inspection of the approximations. Validation was performed on a leave-one-out basis, where all measured data for each subject in turn were eliminated from the training set and subsequently predicted by training data from all other subjects, i.e., no vGRF information from the predicted subject was used in the training data.

The algorithm predicts the Fourier vGRF coefficients from which the predicted vGRF curve can be derived, given the coefficients of the measured microphone signals, the body mass, the running speed and the stance time. This prediction is formulated as a quadratic programming problem in a Principal Component Analysis (PCA)-transformed parameter space that maximizes the likelihood of the predicted vGRF coefficients, given the measured sound coefficients and the statistical variation of the training data (Fourier coefficients of sounds and vGRFs). The result is essentially a conditional likelihood algorithm [20], which works well on normally distributed data.

The vGRF prediction was performed using different sets of microphone combinations, starting from individual microphones (Mic1, Mic2, Mic3 and Mic4) to pairs of microphones representing the posterior sounds (Mic1-Mic4, Mic14), anterior sounds (Mic2-Mic3, Mic23), medial sounds (Mic1-Mic2, Mic12) and lateral sounds (Mic3-Mic4, Mic34). Finally, a prediction using all four microphones was conducted (Mic1234), totaling nine types of vGRF prediction from the recorded sounds.

### 2.6. Data Analysis

In order to attest the vGRF prediction quality in each of the nine microphone combinations, Pearson’s correlation coefficient, absolute root-mean-square error (RMSE) and relative RMSE (rRMSE) were calculated for each real and its respective predicted vGRF curve in each microphone combination. Relevant running biomechanical variables were extracted from both real and predicted individual vGRF curves. The impact peak is a local maximum of the vGRF curve within the first 20% stance [22] usually associated with heel contact with the floor during running. The impact peak, when identifiable, was defined as the local maximum of the vGRF within the first 50 ms following initial contact. Three types of loading rates were calculated: The first (LR-1) was defined as the average slope from initial contact to the impact peak [23]. The second (LR-2) was defined as the slope of the line between 20 and 80% of the impact peak [18]. The third (LR-3) was defined as the slope of the force within the first 50 ms [23].

If the impact peak was not found, an iterative evaluation of the vGRF curve was performed, where the angles between two sequential vectors of 15 ms were calculated from 10 to 100 ms within the vGRF. The time of the highest negative angle (i.e., onset of curve deflection) was defined as the location for the predicted impact peak (see Figure 2 for illustration using both real and predicted vGRF). Figure 2C demonstrates that the method can identify the exact instant of the impact peaks from the real vGRF. The vertical impulse was defined as the area under the vGRF curve. All vGRF parameters were normalized to the participants’ body weight (xBW).

The running style was defined from the presence or absence of the impact peak on the vGRF data. vGRF curves presenting prominent impact peaks were classified as rearfoot running, whereas vGRF curves without the impact peak were classified as forefoot/midfoot running. This procedure was conducted for the real vGRF and for the predicted vGRF extracted from all microphone combinations. Finally, the accuracy in predicting the participants’ running style for each vGRF curve was computed as the number of true positive identifications divided by the total number of predictions.

### 2.7. Statistical Analysis

The normality of all the dependent variables (RMSE, rRMSE, running style accuracy, LR-1, LR-2, LR-3, impact peak, time to impact peak, active peak, time to active peak and vertical impulse) was checked and confirmed using a Kolmogorov–Smirnov test. Group-based differences in RMSE, rRMSE and the accuracy to identify running style across the different vGRF predictions using different microphone combinations were assessed using a 1-way ANOVA for repeated measures. The differences between real and predicted loading rates, impact peak, time to impact peak, active peak, time to active peak and vertical impulse calculations were assessed using paired t-Student tests. Finally, Pearson correlations comparing real vs. predicted biomechanical variables were conducted using all data points from all runners to determine the degree of agreement between the real and predicted variables. Correlations were classified as low (0.1 < r ≤ 0.3), moderate (0.4 < r ≤ 0.7) and strong (r > 0.7) [24]. A significance level of *p* < 0.05 was set for all statistical tests.

## 3. Results

Subjects 1, 4, 5, 7, 9, 11 and 26 had faults in the form of white noise transmissions on one or more microphones and were, therefore, excluded from the study, leaving 37 subjects in the data set. The average running speed was 2.75 ± 0.33 m/s, calculated from 1780 running cycles (48 ± 8 cycles/runner). The final sample size for the investigation was in accordance with recent recommendations of adequate sample sizes for research experiments involving running biomechanics [18]. Each of the 1780 running cycles generated nine different microphone combinations, resulting in a total of 16,020 samples to be predicted. To summarize, the results from each runner, the inter-trial data, were averaged. In general, the quality of the generated vGRF curves across all the microphone combinations was high, with Pearson correlation coefficients exceeding 0.95 with a mean of 0.99 (Figure 3A), absolute RMSE below 0.25 xBW (Figure 3B) and relative RMSE between 5% and 25% (Figure 3C). There were no effects of the microphone combinations neither on the absolute nor relative RMSE (*p* > 0.05).

The vGRF predictions using different combinations of the microphones presented predominantly similar outcomes. Figure 4 illustrates the comparison of the real and reconstructed vGRF across the participants with a high-quality prediction (Figure 4A–D) and lower prediction quality (Figure 4E–H).

### 3.1. Effect of Microphone Combinations on Running Style Prediction

The accuracy for predicting the running style using the vGRF derived from the sounds was conducted by calculating the % of correct classifications in relation to the total number of events for a given runner. The results were subsequently described using the percentage of correctly classified footsteps for each runner. The running styles were classified as rearfoot strike (e.g., presenting an impact peak on the vGRF curve) or forefoot strike (not presenting an impact peak on the cGRF curve).

Similarly, for the results from the absolute and relative RMSE, there was no effect of the microphone combinations on the accuracy to determine whether a runner was a rearfoot or forefoot/midfoot striker (*p* > 0.05). Due to the lack of statistical evidence to define the best microphone combination, we defined that the combination presenting the greatest prediction accuracy against the gold standard from the real vGRF would be used for the further illustration of the results. As evidenced in Table 1, the microphone combination with the greatest foot strike prediction accuracy was Mic34.

### 3.2. Running Style Prediction Using vGRF Generated from Sound Recordings

By using the ankle dorsiflexion angle to characterize running style, we determined that 1241 steps were heel strikes (~70% of the steps), 367 steps were midfoot strikes (~21%) and 172 steps were forefoot strikes (~9%). By using the real vGRF, it was possible to determine that 29 out of the 37 runners presented prominent impact peaks (78% of the sample), whereas only 8 runners (22% of the sample) did not present impact peaks. The use of the predicted vGRF from the sound recordings could accurately predict the presence or absence of impact peaks with an inter-trial accuracy >75% for 22 runners (59% of the sample). From the 22 runners presenting >75% accurate predictions, 16 presented impact peaks and 6 runners did not present impact peaks. The average impact peak prediction accuracy was 77 ± 27% (median = 90%; 95% confidence interval: 68%–86%). Figure 5 illustrates the vGRF variability across all the trials from the selected participants presenting prominent impact peaks (Figure 5A–D) and with absent impact peaks (Figure 5E–H). It is noteworthy that the predictions encountering the impact peaks present greater accuracy when compared to the absent impact peaks.

### 3.3. General Running Biomechanical Variables Extracted from Reconstructed vGRF

The group-based comparison between the biomechanical variables extracted from the real and the predicted vGRF revealed no significant differences in the outcomes (*p* > 0.05, Table 2). However, the effect of the significance is small-to-medium (0.4) and close to a significant difference for the impact peak. The RMSE (computed across individual trials from each runner and averaged to represent the runner) was predominantly fair (<20%), except for LR-2 which presented the greatest error (~15 xBW/s, ~26%), whereas the active peak and vertical impulse presented the lowest errors (<6% on average).

### 3.4. Associations between Real and Predicted vGRF

Pearson correlations were conducted to associate the real and predicted biomechanical variables extracted from the vGRF curves independently. The correlations included data from each footstep of each runner for every biomechanical variable shown in Figure 6, where the gray dots represent individual data points, and the colored dots represent the average of a given participant. There were predominantly moderate-to-strongly significant associations between the real and predicted vGRF across the relevant biomechanical variables (Figure 6). The best-fitted biomechanical variables were the vertical impulse (r = 0.93, Figure 6B) and the active peak (r = 0.69, Figure 6E), whereas the time to active peak (r = 0.18, Figure 6D) and time to impact peak (Figure 6B) presented the weakest associations. Moreover, LR-1 and LR-3 presented greater correlation between the real and predicted calculations (r = 0.62) when compared to LR-2 (r = 0.50).

## 4. Discussion

The main findings of the present study were that there is no statistical difference in the quality of the vGRF predictions using single, pairs or all recorded sound sources, and the use of the predicted vGRF to determine running style using the vGRF impact peak (heel strike vs. forefoot strike) provided a mean accuracy of ~77%. In addition, no statistical differences were found between the running biomechanical variables extracted from the real and predicted vGRF, whereas the association between the real and predicted running biomechanical variables was moderate to high (r = 0.93 for vertical impulse). Taken together, our results suggest that the use of sound waves is a feasible way to reconstruct the vGRF and extract relevant biomechanical variables during running. Our results open new avenues for the exploration of methods to investigate running biomechanics in outdoor settings.

The inclusion of sounds from multiple microphones did not provide a superior prediction of the vGRF when compared to one or pairs of microphones. Determining that one microphone is enough to provide useful data for successful GRF reconstruction indicates that a simple methodology using one microphone may suffice for data recordings. Interestingly, the combination with the greatest accuracy in predicting a forefoot or rearfoot strike was Mic34, consisting of microphones located laterally to the runner. A previous study using the same microphone setup demonstrated moderate correlations between the sounds of frontal microphones to the forces generated by runners during soft impact, whereas forces from the posterior microphones were correlated with the forces generated by running in more natural impact conditions [12]. It is plausible that the use of multiple microphones for the description of only self-selected running might insert redundant information across multiple microphone sources, despite the possible difference in sound properties generated across different running styles. Therefore, our study contributes to the field by proposing that lateral microphones may offer optimal data to acquire input data for predicting the vGRF from sounds.

Our results indicate that the prediction of the vGRF from the sounds of footsteps during running is feasible and could be relevant for the assessment of biomechanical running parameters for improvements in performance as well as injury-risk assessment. Such advances would be possible via inexpensive sensors in the form of shoe-mounted microphones. However, the prediction of vGRF quality is variable-dependent, despite an overall high association between the real and predicted vGRF curves. Pearson correlations are insensitive to specific features of curves, such as the impact peaks, providing an overestimated assessment of the reconstruction quality. By assessing the root-mean-square errors, we found deviations between 5 and 20%, which underpins considerable differences between the curves, especially around the impact peak location. However, there is no tangible literature to compare our results with, since this is the first study using sounds to reconstruct the vGRF.

Previous studies reconstructing the vGRF demonstrated lower errors of ~<5% or ~<50 N during walking [25,26] and running [4,27]. Interestingly, some studies demonstrated lower prediction errors from basic running features, such as the active peak and vertical impulse [2,25], but either did not compute the impact peak [2] or did not provide an evaluation of the impact peak present in the predicted curves [28]. Pogson and co-workers [27] predicted the ground reaction force curves from torso-mounted accelerometers and found a relative RMSE >0.2 kN or >12% of the peak forces investigated. When evaluating specific variables, the authors did not compute the impact parameters from curves that did not include a clear impact peak, therefore reducing their total number of predictions from 164 running impacts to fewer than 90 impacts. Nonetheless, the authors found a moderate-to-high association between the real and predicted impact peaks (r^2^ = 0.74) and loading rates (r^2^ = 0.63). Defining the impact peaks is essential to identifying the loading rates and evaluating the success of gait retraining protocols. Our results demonstrate slightly lower associations for the impact peaks and loading rates, but the parameters were extracted from all 1241 running impacts with or without prominent impact peaks. Therefore, our results demonstrate a clear advance in detecting the impact properties for vGRF curves that do not possess impact peaks, which could be accurately identified in more than 80% of all cases.

Despite the apparently good predictions related to impact peaks, the biomechanical variables with the highest prediction quality were the active peak and the vertical impulse (Figure 5). Neither of these variables depend on the prediction of other features and are highly related to the fundamental principles of center-of-mass acceleration throughout the running stance. Previous studies also found a high prediction quality for the active peak and vertical impulse [28,29]. On the other hand, the prediction of the loading rates—which can be dependent on the identification of an impact peak on the vGRF curve—presented a lower prediction quality. In fact, our loading rate prediction, based on the force slope from 20% to 80% of the impact peak (LR-2), presented a substantially greater error (~20%) when compared to LR-1, which is calculated from the force slope within the first 50 ms (~11%). Therefore, further development of the Fourier transform algorithm may be necessary to reduce the error related to the identification of the impact peaks.

A recent systematic review with meta-analysis [6] found no conclusive evidence supporting that biomechanical and musculoskeletal measurements can be considered risk factors for running-related injuries. Therefore, studies conducted with an adequate sample size using inexpensive techniques, such as sound recordings, may be a relevant alternative to strengthen the relevance of biomechanical analysis for injury prevention. The implementation of shoe-mounted sensors, such as microphones or other miniaturized technologies, is a natural step from this current work, which can provide ideal data acquisition possibilities to perform short- and long-term monitoring of runners throughout their training sessions and competitions. Moreover, the proposed method can inform runners about biomechanical parameters and can add kinetic information to the mostly kinematic information available from inertial measurement units and other sensor types typically available on the market today.

It is noteworthy that the floor in the laboratory was a smooth, hard and painted surface, and the influence of different sounds from different surface types remains to be investigated. Moreover, participants used the same shoe model, eliminating potential sound variability caused by runners using different shoe models. It is likely that the surface properties and shoe models influence the recorded sounds to such an extent that machine learning algorithms must be trained with sounds from a variety of different surfaces and shoes. Therefore, our prediction results are limited to the specific floor–shoe interaction conditions of our experimental design.

## 5. Conclusions

In summary, our results demonstrated the feasibility of using acoustic microphones in combination with machine learning to predict vertical ground reaction force curves, with no apparent advantage in using several microphones for the predictions. We found a fair mean accuracy (~77%) in determining running patterns by using the presence/absence of impact peaks, whereas the prediction of relevant running biomechanical features presented a moderate-to-high association with the real features.

## Figures and Tables

**Figure 1 sensors-22-09640-f001:**
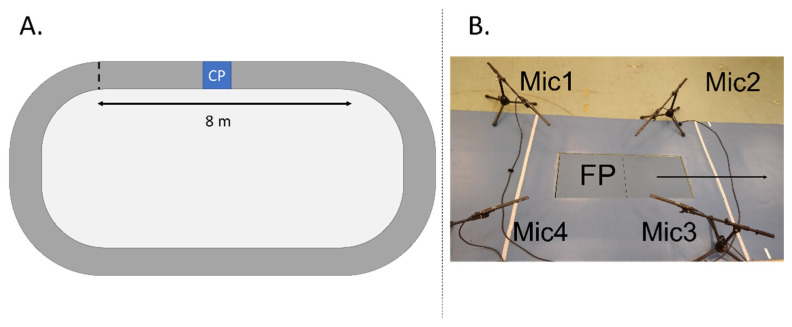
(**A**) Illustrative running course with a straight running line of 8 m and a caption point (CP) in the middle; (**B**) Up-close illustration of the caption point with the positioning of the microphones (Mic1-Mic4) and the force plate (FP), where the arrow indicates the running direction.

**Figure 2 sensors-22-09640-f002:**
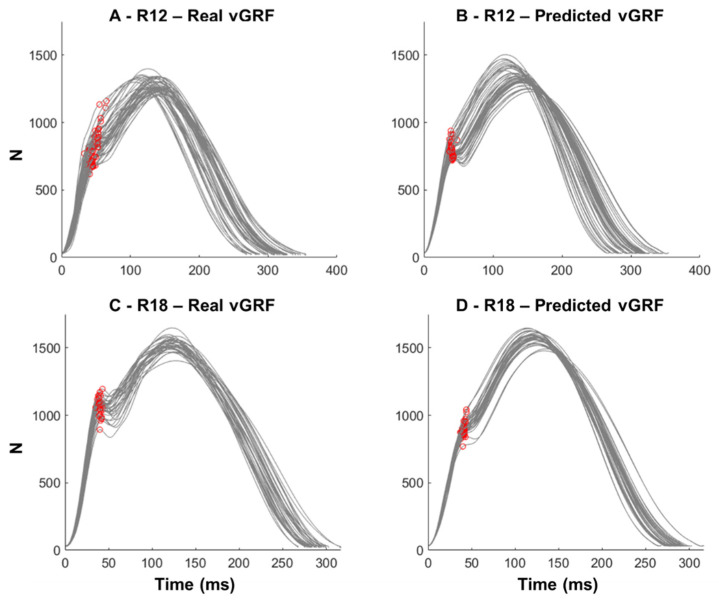
Illustrative examples from two runners (Runner 12 in plots (**A**) and (**B**); Runner 18 in plots (**C**) and (**D**)) of the identification of impact peak instants from real and predicted vertical ground reaction forces (vGRF, in Newton (N)) in runners not presenting prominent impact peak. The red circles identify the transition point where the vGRF curve presents a deflection.

**Figure 3 sensors-22-09640-f003:**
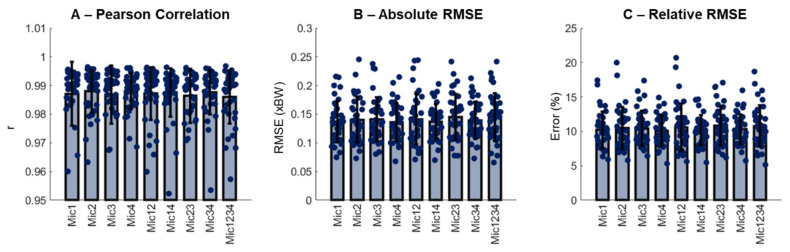
Mean (SD) and data distribution from the study sample (N = 37) of Pearson correlation coefficient (r, panel (**A**)), absolute root-mean-square error (RMSE, panel (**B**)), and relative RMSE (panel (**C**)).

**Figure 4 sensors-22-09640-f004:**
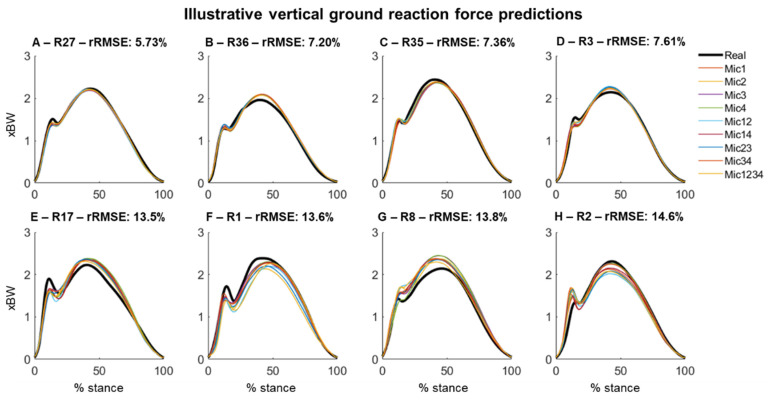
Vertical ground reaction force prediction (normalized to body weight [xBW]) using sounds recorded from different combinations of microphones. Each curve is an average across all cycles from the same runner. The top row (**A**–**D**) shows the data for selected runners (R) with the lowest average relative root-mean-square prediction error (rRMSE), whereas the bottom row (**E**–**H**) shows the data for runners with the highest prediction error.

**Figure 5 sensors-22-09640-f005:**
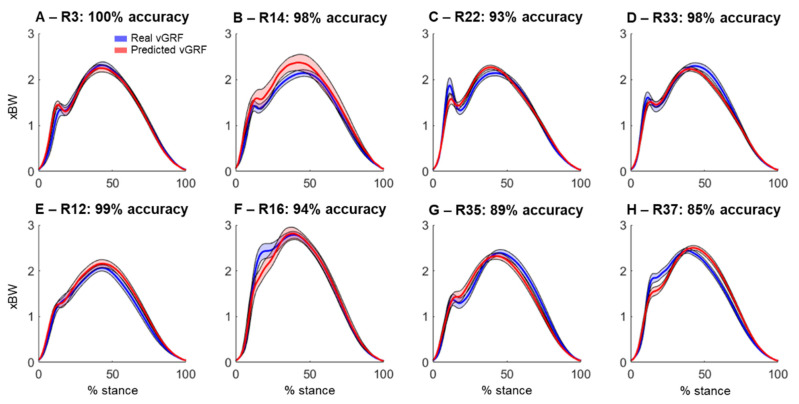
Real (blue) and predicted (red) vertical ground reaction forces (vGRF) from selected runners (R) presenting prominent impact peaks (**A**–**D**) and with absent impact peaks (**E**–**H**). Thick lines represent data average and shaded areas represent standard deviation. For each runner, the overall accuracy in determining presence/absence of impact peaks is shown in the figure subplot titles.

**Figure 6 sensors-22-09640-f006:**
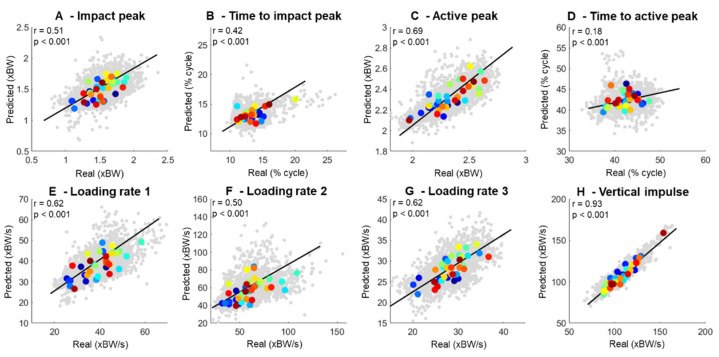
Association between running biomechanical variables extracted from 1716 running trials (*gray points in the background*) using real vs. predicted vertical ground reaction forces. Each participant’s average across all trials (N = 37) is represented as the same large colored dot across all subplots.

**Table 1 sensors-22-09640-t001:** Descriptive statistics from the accuracy in predicting running style using vertical ground reaction forces generated from sound data (% of total predictions). SD = standard deviation; CI-lower = lower bound of the 95% confidence interval; CI-upper = upper bound of the 95% confidence interval.

Microphone	Mean (SD)	Median	CI-Lower	CI-Upper
Mic1 (%)	73.02 ± 30.9	86.93	62.69	83.34
Mic2 (%)	69.79 ± 31.0	79.62	59.43	80.15
Mic3 (%)	74.42 ± 27.4	81.86	65.28	83.55
Mic4 (%)	70.76 ± 33.5	85.79	59.59	81.94
Mic12 (%)	73.27 ± 29.0	86.81	63.60	82.94
Mic14 (%)	74.51 ± 25.7	76.39	65.94	83.08
Mic23 (%)	72.07 ± 27.1	75.87	63.02	81.12
Mic34 (%)	77.10 ± 26.74	89.73	68.18	86.01
Mic1234 (%)	74.95 ± 24.8	79.13	66.66	83.23

**Table 2 sensors-22-09640-t002:** Comparison between running biomechanical variables extracted from the real vertical ground reaction forces (vGRF) and the predicted GRF. T2 = time to; LR = loading rate; BW = body weight; rRMSE = relative root-mean-square error.

Running Variables	Real vGRF	Pred vGRF	*p*-Value	Cohen’s d	RMSE
Impact peak (xBW)	1.51 ± 0.20	1.48 ± 0.15	0.25	0.16	0.22 ± 0.06
T2 impact peak (% cycle)	13.51 ± 0.71	13.39 ± 0.93	0.61	0.08	1.56 ± 0.87
Active peak (xBW)	2.32 ± 0.17	2.31 ± 0.13	0.63	0.05	0.12 ± 0.04
T2 active peak (% cycle)	42.33 ± 2.41	42.37 ± 1.63	0.91	0.02	3.16 ± 1.46
LR-1 (xBW/s)	39.30 ± 7.69	38.27 ± 5.92	0.26	0.15	6.69 ± 2.62
LR-2 (xBW/s)	59.65 ± 17.14	57.32 ± 12.21	0.33	0.15	15.46 ± 8.91
LR-3 (xBW/s)	28.35 ± 3.56	28.15 ± 3.04	0.62	0.05	3.44 ± 1.06
Vertical impulse (xBW/s)	107.80 ± 14.5	107.52 ± 14.9	0.98	0.00	5.21 ± 2.39

## Data Availability

Raw data are available from the authors upon request.

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
