# Peer review of "Predicting Vertical Ground Reaction Forces in Running from the Sound of Footsteps"

_sensors, 2022, doi:10.3390/s22249640_

Round 1

Reviewer 1 Report

a paragraph with the overal description of the existing sections is missing. It should be added in the Introuction after line 76. 

There is an inconsistency regarding the sampling size of the experiment. On the abstract (line 13), it is mentioned that 16020 footsteps from 37 runners were processed, whereas in line 221 that there are 16020 predicted samples. The latter statement implies that 1780 footsteps (or running cycles) were processed across 9 microphone combinations totalling 16020 predictions. The authors should be more concise and clearer with the sampling size and absolute number of total impacts analysed.

In line 306, it is concluded that a median accuracy of 89% was achieved on determining the running style of a participant using the impact peak, derived from the predicted vGRF. However, in line 265 it is stated that while real vGRF data allowed an impact or no impact verdict for all valid participants, for the predicted vGRF a conclusion could be drawn for only 22 out of 37. Accuracy claims should not be formed while excluding 41% of the sampling size.

 In line 220, it is mentioned that the final sample size is in accordance with recommendations of adequate sample sizes for experiments involving running biomechanics. According to the reference 19, the proposed size of this work holds true from a data stability and statistical significance point of view. However, nothing is mentioned on the sample size in regards to the machine learning technique used. Literature should be considered.

For line 153, authors should elaborate on the criteria regarding the chosen number of Fourier coefficients.

Author Response

We thank the reviewer very much for the constructive comments to our manuscript.

A paragraph with the overal description of the existing sections is missing. It should be added in the Introuction after line 76. 

R: A paragraph with the overall outline of the paper has been added at the end of the introduction.

There is an inconsistency regarding the sampling size of the experiment. On the abstract (line 13), it is mentioned that 16020 footsteps from 37 runners were processed, whereas in line 221 that there are 16020 predicted samples. The latter statement implies that 1780 footsteps (or running cycles) were processed across 9 microphone combinations totalling 16020 predictions. The authors should be more concise and clearer with the sampling size and absolute number of total impacts analysed.

R: We acquired 1780 trials from 37 runners, which contained four microphone data and vertical ground reaction force data. Since we generated nine different microphone data combinations, we conducted 16020 predictions (1780 x 9 = 16020). The text referring to the total amount of predicted footsteps was removed from the abstract in the interest of clarity, and we improved the description in the first paragraph of the results section, lines 311-312.

In line 306, it is concluded that a median accuracy of 89% was achieved on determining the running style of a participant using the impact peak, derived from the predicted vGRF. However, in line 265 it is stated that while real vGRF data allowed an impact or no impact verdict for all valid participants, for the predicted vGRF a conclusion could be drawn for only 22 out of 37. Accuracy claims should not be formed while excluding 41% of the sampling size.

R: We removed the mentioning of the median accuracy from the results, maintaining the mean results of 77% mean prediction accuracy across all participants. However, the expressed results are not excluding participants. The 77% mean accuracy includes all 37 runners, whereas some runners have greater accuracy than others. In addition, our 95% confidence interval shows that the lower bound of the 95-CI is 68%, meaning that our worse participants present fair number of accurate reconstructions. Therefore, we believe that a mean accuracy of 77% is the correct way to state our results.

In line 220, it is mentioned that the final sample size is in accordance with recommendations of adequate sample sizes for experiments involving running biomechanics. According to the reference 19, the proposed size of this work holds true from a data stability and statistical significance point of view. However, nothing is mentioned on the sample size in regards to the machine learning technique used. Literature should be considered.

R: This is  a very valid point. The mentioning of the previous literature is relevant due to the use of biomechanical parameters such as loading rates and vertical impulses. Concerning machine learning, there are no guidelines expressing the minimum sample size, and the authors believe that the total number of predictable events is more relevant than the runner’s sample size. This is especially relevant since predictions are performed for individual events (in this case footsteps) and whether a runner presents 10 or 100 steps does not make a difference for the prediction. However, it does make a difference for the training of the model. The studies we cite in our work have a maximum of 18 runners (Honert et al. 2022), demonstrating that our sample size is superior to the correspondent literature.

For line 153, authors should elaborate on the criteria regarding the chosen number of Fourier coefficients.

R: This section has been extended with more precise information and a reference to previous work on the matter.

Reviewer 2 Report

This paper uses an unspecified machine learning approach to assess ground reaction forces in running using information from runners such as microphone signals FFT coefficients, the body mass, the running speed and the angular frequency of runners.

For perfection, I strongly suggest authors consider the following suggestions/comments:

-       Is the dataset available online?

-       A picture of the equipment over a runner should be included in the paper.

-       The machine learning approach should be described. Explain why this machine learning technique is used in this particular case.

-       Clarify the difference between LR-1 and LR-3.

-       Why the lack of impact peak happened?

-       Include a Figure to observe impact peak in order to compare to Figure 1 situations.

-       It would be interesting that the different classification and regression tasks indicated in the paper were deeply described and indicated in each part of the paper.

-       It is said that there is no significant difference between using 1 microphone or a combination of several microphones. Then, what is the contribution of the deep analysis?

-       How many footsteps are you managing without considering the microphone combinations?

-       Section 3.1 indicates that “the microphone combination with the greatest foot strike prediction accuracy was Mic34.” This is true. However, there is no statistical difference. It should be indicated. Why this analysis is performed if there is no difference?

-       Section 3.2 indicates that “From the 22 accurate predictions, 16 presented impact peaks”. However, if only 16 presented impact peaks, then the remaining 6 (22-16) are not “accurate predictions”.

-       I think that it is not fair to highlight the median accuracy instead of the mean accuracy in the discussion section.

Author Response

The authors thank the reviewer for the constructive comments to our manuscript.

This paper uses an unspecified machine learning approach to assess ground reaction forces in running using information from runners such as microphone signals FFT coefficients, the body mass, the running speed and the angular frequency of runners.

For perfection, I strongly suggest authors consider the following suggestions/comments:

Is the dataset available online?

The dataset is not available online, but it will be made available to readers upon request. The Data Availability Statement has been updated accordingly.

A picture of the equipment over a runner should be included in the paper.

A new figure 1 outlining the lab setup has been added to the manuscript.

The machine learning approach should be described. Explain why this machine learning technique is used in this particular case.

We are unsure whether the reviewer may have missed that the machine learning was described including mathematical derivations in detail in a previously published and open access publication, which is reference #21 in the new version of the paper. In the interest of completeness, we have added additional verbal information about the about the workings of the algorithm in in lines 210 through 226 in the revised manuscript. We have also added the information that this type of algorithm works well for normally distributed data. We believe that the results document that it does indeed work well for the present case.

Clarify the difference between LR-1 and LR-3.

Additions have been made to Section 2.6, which described the three different types of loading rates. The difference between LR-1 and LR-3 is that LR-1 is the slope from the touch-down to the impact peak, whenever it occurs in time (variable time period. On the other hand, LR-3 is the slope from touch-down to the force 50 ms after touch-down (fixed time period)

Why the lack of impact peak happened?

Impact peaks represent the early instantaneous loading during touch-down, usually associated with heel contact with the floor during running. The lack of impact peak events suggests a smoother touch-down event, usually associated with forefoot striking. However, this is not a consensus in the literature (https://pubmed.ncbi.nlm.nih.gov/24300124/). We included the definition of impact peak in the methods (section 2.6).

Include a Figure to observe impact peak in order to compare to Figure 1 situations.

The example on Figure 2C exemplifies the existence of impact peaks. This information has been added to the text:

“Figure 2C demonstrates that the method can identify the exact instant of the impact peaks from the real vGRF.”

Figures 4 and 5 also provide examples of vertical ground reaction force curves presenting impact peaks. Therefore, the authors believe it is not necessary to include another figure in the manuscript.

It would be interesting that the different classification and regression tasks indicated in the paper were deeply described and indicated in each part of the paper.

We added short descriptions of the methodologies for the calculation of running style prediction (section 3.1) and Pearson correlations associating real and predicted biomechanical variables (section 3.4)

It is said that there is no significant difference between using 1 microphone or a combination of several microphones. Then, what is the contribution of the deep analysis?

The deep analysis was relevant to establish that the use of multiple microphones does not improve the quality of the ground reaction force reconstruction, since only one microphone provides data for reconstructing the GRF curve and predict the presence/absence of impact peaks with fair-to-good accuracy. This is relevant for the prospect of establishing a wearable sensing technology based on the findings. We included the following sentence in the discussion:

“Determining that one microphone is enough to provide useful data for successful GRF reconstruction is highly relevant, since it established that a simple methodology using one microphone may suffice the need for data recordings.”

How many footsteps are you managing without considering the microphone combinations?

We recorded 1780 footsteps with four microphones and GRF data, without considering any combination. This information is now elaborated upon in the first paragraph of the Results section.

Section 3.1 indicates that “the microphone combination with the greatest foot strike prediction accuracy was Mic34.” This is true. However, there is no statistical difference. It should be indicated. Why this analysis is performed if there is no difference?

The lack of differences between the microphone combinations is a relevant information to establish future methodologies to further develop the proposed method, since one microphone is enough to provide relevant data. Describing the results from all nine combinations from section 3.2 forward in the manuscript would not be reasonable, therefore we established the criteria based on the best performing (even though not significant) combination to define which combination to report.

Section 3.2 indicates that “From the 22 accurate predictions, 16 presented impact peaks”. However, if only 16 presented impact peaks, then the remaining 6 (22-16) are not “accurate predictions”.

The authors believe the interpretation of the reviewer on the results is due to a lack of clarity in our writing. From a total of 22 runners, 16 presented predominance of impact peak occurrence in their ground reaction force. However, that does not mean that the other six runners were not accurately predicted. The other six runners presented predominance of no impact peak occurrence, and this lack of impact peak occurrence was also correctly predicted, since the predicted ground reaction force curves did not present impact peaks for these runners.

The sentence has been corrected as follow:

“The use of predicted vGRF from sound recordings could accurately predict the presence or absence of impact peaks with an inter-trial accuracy >75% for 22 runners (59% of the sample). From the 22 runners presenting >75% accurate predictions, 16 presented impact peaks and 6 runners did not present impact peaks.”

I think that it is not fair to highlight the median accuracy instead of the mean accuracy in the discussion section.

The authors removed the mentioning of the median accuracy and highlighted the mean accuracy in the discussion.

Round 2

Reviewer 2 Report

The authors have addressed the comments of previous review.

I suggest correcting some typos:

-       Include a , when number 1,780 appears.

-       I think that the median specified in line 305 should be 89% instead of 88% according to Table 1.

Author Response

We once again thank the reviewer for the effort, which has really been most helpful in terms of improving the quality of the paper.

  • Include a , when number 1,780 appears
    Done
  • I think that the median specified in line 305 should be 89% instead of 88% according to Table 1.
    You are quite right, and there were actually a couple of other numbers in the text that did not cite the table correctly. This has all been fixed.